# Dissecting the role of PfAP2-G in malaria gametocytogenesis

Gabrielle A. Josling[1,2], Timothy J. Russell [1,2], Jarrett Venezia[1,2,4], Lindsey Orchard [1,2], Riëtte van Biljon [1,2], Heather J. Painter[1,2,5] & Manuel Llinás[1,2,3 ✉]

In the malaria parasite *Plasmodium falciparum*, the switch from asexual multiplication to sexual differentiation into gametocytes is essential for transmission to mosquitos. The transcription factor PfAP2-G is a key determinant of sexual commitment that orchestrates this crucial cell fate decision. Here we identify the direct targets of PfAP2-G and demonstrate that it dynamically binds hundreds of sites across the genome. We find that PfAP2-G is a transcriptional activator of early gametocyte genes, and identify differences in PfAP2-G occupancy between gametocytes derived via next-cycle and same-cycle conversion. Our data implicate PfAP2-G not only as a transcriptional activator of gametocyte genes, but also as a potential regulator of genes important for red blood cell invasion. We also find that regulation by PfAP2-G requires interaction with a second transcription factor, PfAP2-I. These results clarify the functional role of PfAP2-G during sexual commitment and early gametocytogenesis.

[1] Department of Biochemistry and Molecular Biology, The Pennsylvania State University, University Park, PA, USA. [2] Huck Center for Malaria Research, The Pennsylvania State University, University Park, PA, USA. [3] Department of Chemistry, The Pennsylvania State University, University Park, PA, USA. [4]Present address: W. Harry Feinstone Department of Molecular Microbiology and Immunology, Bloomberg School of Public Health, Johns Hopkins University, Baltimore, MD, USA. [5]Present address: Division of Bacterial, Parasitic, and Allergenic Products, Office of Vaccines Research and Review, Center for Biologics Evaluations and Research, Food and Drug Administration, Silver Spring, MD, USA. ✉email: manuel@psu.edu

Though progress is being made toward the eradication of malaria, the disease continues to be a major global public health burden. Nearly half the world's population is at risk of malaria and hundreds of millions of people suffer from the disease each year[1]. The malaria parasite *Plasmodium falciparum* has a complex lifecycle that involves development in the mosquito as well as the liver and red blood cells of humans. The symptoms of the disease are associated only with the 48-h intraerythrocytic cycle, during which the parasite undergoes repeated rounds of multiplication, egress, and reinvasion as it passes through the ring, trophozoite, and schizont stages. However, these asexual stages cannot be productively taken up by the mosquito and transmitted. Instead, the parasite must undergo sexual differentiation to form male and female gametocytes. Crucially, only a small proportion of cells in each asexual cycle will commit. Commitment to sexual development generally occurs prior to schizogony, such that every merozoite within a single schizont will either enter sexual development or continue asexual multiplication following egress and reinvasion—this is known as next-cycle conversion and seems to be the canonical pathway[2]. However, it is also possible for commitment to occur in the very early ring stage resulting in parasites developing directly into gametocytes without first passing through schizogony (same-cycle conversion)[3]. During the 10–12 day process of gametocytogenesis, the parasite undergoes dramatic morphological changes as it passes through five distinct stages (stages I–V)[4,5]. With the exception of differentiated stage V forms, all other gametocyte stages are sequestered in host tissues, particularly the bone marrow[6,7].

One critical regulator of sexual commitment is AP2-G, which is a member of the ApiAP2 family of DNA-binding proteins[8,9]. Forward and reverse genetics in *P. falciparum*, *Plasmodium berghei*, and *Plasmodium yoelii* has demonstrated that AP2-G is the master regulator of gametocytogenesis[8–12]. Parasites that do not express *ap2-g* will continue asexual development, whereas those that do are able to commit and enter gametocytogenesis. In *P. falciparum*, *ap2-g* is not expressed in most cells because it is epigenetically silenced by H3K9me3, heterochromatin protein 1 (HP1), and histone deacetylase 2 (Hda2)[13–15]. HP1-mediated silencing of *ap2-g* is a ubiquitous feature of its regulation as it has recently been demonstrated across a range of *Plasmodium* species[16]. In *P. falciparum*, the protein GDV1 is responsible for removing HP1 from the *ap2-g* locus and so is also an important positive regulator of commitment[17–19]. Although some aspects of the regulation of PfAP2-G itself have now been explored, relatively little is known about how PfAP2-G directs the sexual commitment transcriptional program and which genes are involved. Furthermore, although PfAP2-G evidently plays a key role in regulating the commitment phase, its role in later stages of gametocytogenesis is unknown.

PfAP2-G is able to bind to a specific DNA motif in vitro that is enriched within the promoters of gametocyte genes and it is able to activate expression of reporter genes, suggesting that it is an activator of transcription[8]. A recent study using single cell RNA-seq (scRNA-seq) revealed that PfAP2-G+ schizonts are chiefly characterized by a small number of upregulated transcripts and very few that are downregulated relative to asexual schizonts, which is consistent with this model[20]. Although collectively these data suggest that PfAP2-G likely directs commitment by binding the promoters of gametocyte genes and activating transcription, studies to date do not distinguish between direct and indirect effects of PfAP2-G in vivo. In addition, recent work indicates PfAP2-G may play a role in directing transcription beyond commitment, as it is present in the nucleus until stage I of gametocyte development[3]. Identification of the direct targets of PfAP2-G is thus crucial for understanding how commitment is

regulated and will provide insight into the earliest events occurring during gametocytogenesis and define markers of sexual commitment.

In this study, we use the complementary approaches of chromatin immunoprecipitation followed by next generation sequencing (ChIP-seq) and transcriptomic characterization of a conditional knockdown cell line to identify targets of the transcription factor PfAP2-G. We find that PfAP2-G is not only associated with many known and putative early gametocyte gene promoters in the committed schizont phase, but also in sexual rings and stage I gametocytes. Surprisingly, we find that PfAP2-G is also associated with the promoters of a number of genes that are primarily associated with asexual development. Further, stage I gametocytes produced via the two known conversion pathways (NCC and SCC) have many differences in PfAP2-G occupancy, suggesting that PfAP2-G plays different roles in the two pathways. We show conclusively that PfAP2-G is a transcriptional activator of early gametocyte genes both by linking direct binding of PfAP2-G to increased transcript levels and by using CRISPR/Cas9 to mutate putative binding sites upstream of *ap2-g* itself. Interestingly, we find that PfAP2-G interacts with the known transcriptional activator PfAP2-I[21] to regulate transcription during sexual commitment.

## Results

**PfAP2-G is associated with hundreds of genes during gametocytogenesis.** To identify sites in the *P. falciparum* genome that are directly bound by PfAP2-G, we performed chromatin immunoprecipitation followed by next generation sequencing in the previously generated AP2-G-DD parasite line[8]. In this line, PfAP2-G is endogenously tagged with an FKBP destablization domain (ddFKBP). This line produces gametocytes only in the presence of Shld1, though as in wild-type parasites only a subpopulation of cells commit in each cycle (~20%)[8]. ChIP-seq was performed at three stages: committed schizonts, sexual rings, and stage I gametocytes (Supplementary Fig. 1a, b). Because parasites were grown in the presence of Shld1 for the entire duration of the experiment, cells could potentially commit either at the end of cycle 1, the end of cycle 2, or the beginning of cycle 2 (if parasites undergo same-cycle conversion as recently described[3]). For simplicity we refer to parasites collected at the end of the second cycle as stage I gametocytes, though it is possible that this population consists of a combination of committed schizonts and stage I gametocytes produced via both same-cycle conversion and next-cycle conversion.

Our results show that binding of PfAP2-G is dynamic, with hundreds of binding sites across the three stages including many that are stage specific (Fig. 1a, Supplementary Datas 1–4). We identified 195 PfAP2-G peaks in schizonts, 167 in rings, and 404 in stage I gametocytes (Fig. 1b). The vast majority of binding sites for each stage were in intergenic regions and most fell upstream of at least one gene. The chromosomal regions bound by PfAP2-G in all three stages were highly enriched in a DNA sequence motif very similar to that bound by PfAP2-G in vitro[22], confirming the success of the ChIP (Fig. 1c, Supplementary Fig. 2).

We found that many known early gametocyte markers (such as *pfs16*, *etramp10.3*, *gexp05*, *pfg14.744*, and *pfg14.748*) were associated with PfAP2-G binding in stage I gametocytes but not in the earlier stages. Because most previous work has focused on the role of PfAP2-G during commitment we were surprised to see that the protein has the most binding sites in stage I gametocytes. This is, however, consistent with recent findings that PfAP2-G is required until at least the sexual ring stage and is present in the nucleus in stage I gametocytes[3]. While this does not exclude a role

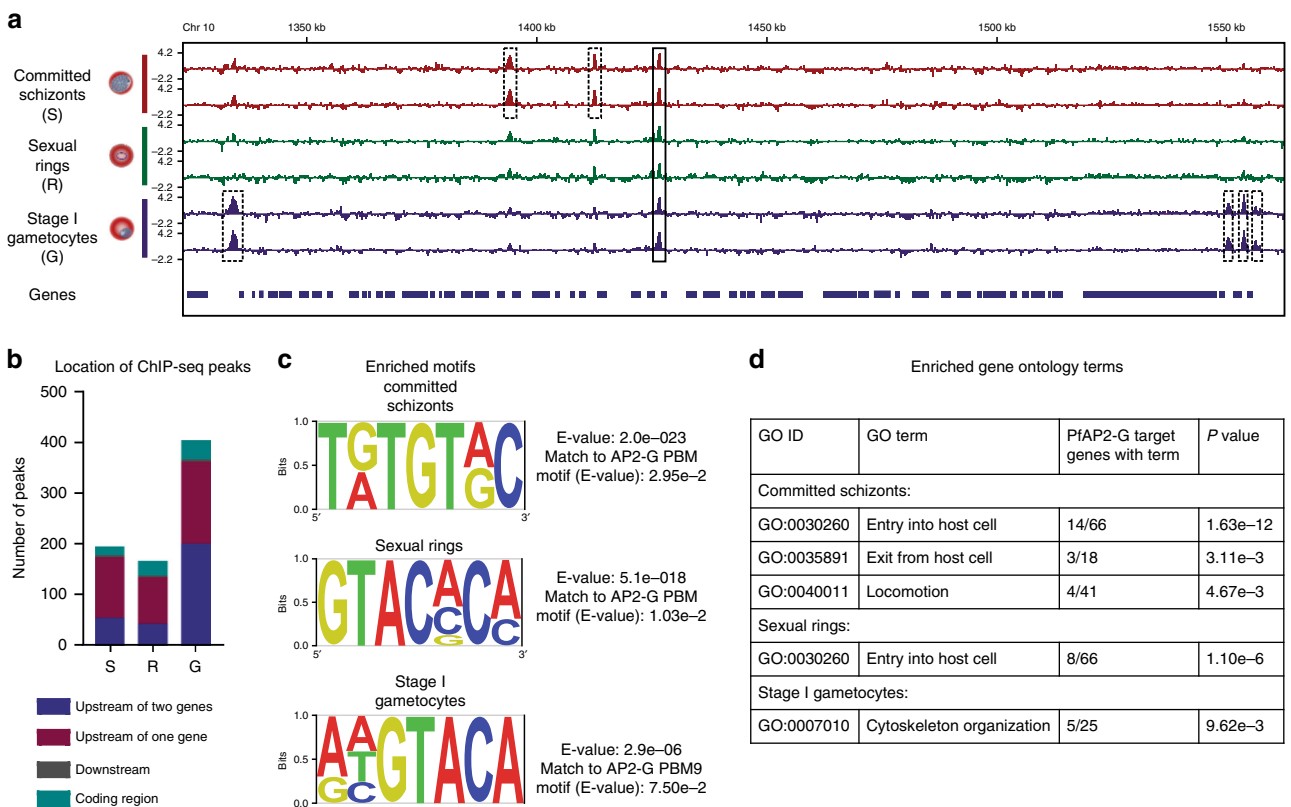

**Fig. 1 PfAP2-G dynamically binds hundreds of sites across the genome throughout commitment and early gametocytogenesis. a** Log2-transformed PfAP2-G ChIP/input ratio tracks for committed schizonts (red), sexual rings (green), and stage I gametocytes (purple) over a region of chromosome 10. Dashed boxes highlight stage-specific binding sites and solid boxes indicate common binding sites. The blue bars underneath show the locations of genes. Data are shown for each of two biological replicates per stage. **b** Bar plot showing the number of PfAP2-G binding sites identified in each stage using ChIP-seq, with the location of each peak indicated by colour. **c** DNA motif analysis of ChIP-seq peaks using DREME identified GTRC (committed schizonts) and GTAC (sexual rings and stage I gametocytes) as the top-scoring motifs. Comparison of each motif to known DNA motifs bound by recombinant AP2 domains using Tomtom showed that all three motifs match most closely to the DNA motif bound by the AP2 domain of PfAP2-G. **d** Gene ontology analysis of the closest genes within 2kb of a PfAP2-G binding site for each stage.

for other transcriptional regulators during early gametocytogenesis, our results indicate that PfAP2-G likely continues to play a critical role in directing gametocytogenesis beyond commitment into the early stages of gametocyte development.

Though the finding that PfAP2-G regulates early gametocyte genes was expected, we show that many genes previously predicted to be controlled by PfAP2-G binding to the promoter (based on transcriptomic studies[8,12,20]) are likely not direct targets of PfAP2-G (Supplementary Data 5). Of particular note, of the 27 genes upregulated in PfAP2-G+ AP2-G-DD schizonts identified using scRNA-seq[20], only 13 are directly bound by PfAP2-G in schizonts (though 7 more are bound in later stages). Similarly, only four of the 23 genes that are down-regulated in lines in which *pfap2-g* has been disrupted are directly bound by PfAP2-G in any stage[8]. This implies that many transcriptional consequences of PfAP2-G induction are due to other as-yet-unidentified downstream regulators and highlights the importance of ChIP-seq in discriminating direct and indirect effects of transcription factors. We also found that PfAP2-G binds the promoters of genes that are differentially transcribed in both male and female gametocytes with no clear bias in either direction, though genes that encode products which can ultimately be detected at the protein level in females (rather than translationally repressed) are overrepresented among PfAP2-G targets[23] (Supplementary Data 5). In addition, of the 49 genes upregulated in the 6 h following overexpression of *P. berghei* AP2-G (PbAP2-G) that have orthologues in *P. falciparum*, only 8 are direct targets of

PfAP2-G[12] (Supplementary Data 5). This finding points to very significant species-specific differences in AP2-G target genes.

We also identified many additional PfAP2-G target genes. Surprisingly, PfAP2-G was associated with the promoters of several upsB *var* genes that encode PfEMP1 proteins that are important in immune evasion and cytoadhesion (Supplementary Datas 2–4). Gene ontology analyses also revealed that genes associated with PfAP2-G binding in committed schizonts and sexual rings were highly enriched in functions related to red blood cell invasion (Fig. 1d). This finding was unexpected, as invasion is a process that occurs specifically in the asexual stages and not in gametocytes. After a sexually committed merozoite invades a red blood cell, the parasite undergoes gametocytogenesis within the same cell and thus does not need to reinvade. Although some invasion proteins also play a role in late-stage gametocytes[24], to our knowledge this is not true of most. Invasion is an essential and extremely complex process involving many proteins that are located in multiple compartments of the cell (particularly the rhoptries, micronemes, and merozoite surface)[25]. PfAP2-G was enriched in the promoters of genes belonging to most of the major families of invasion genes: *msps*, *seras*, *rons*, *raps*, *ebas*, *rhs*, *gaps*, etc. (Supplementary Fig. 3a). Although PfAP2-G is found in the promoters of invasion genes in all three stages, the highest degree of enrichment occurs in the schizont stage (Supplementary Fig. 3b).

Overall, these data demonstrate that PfAP2-G is a direct regulator of early gametocyte genes and thereby acts as the key

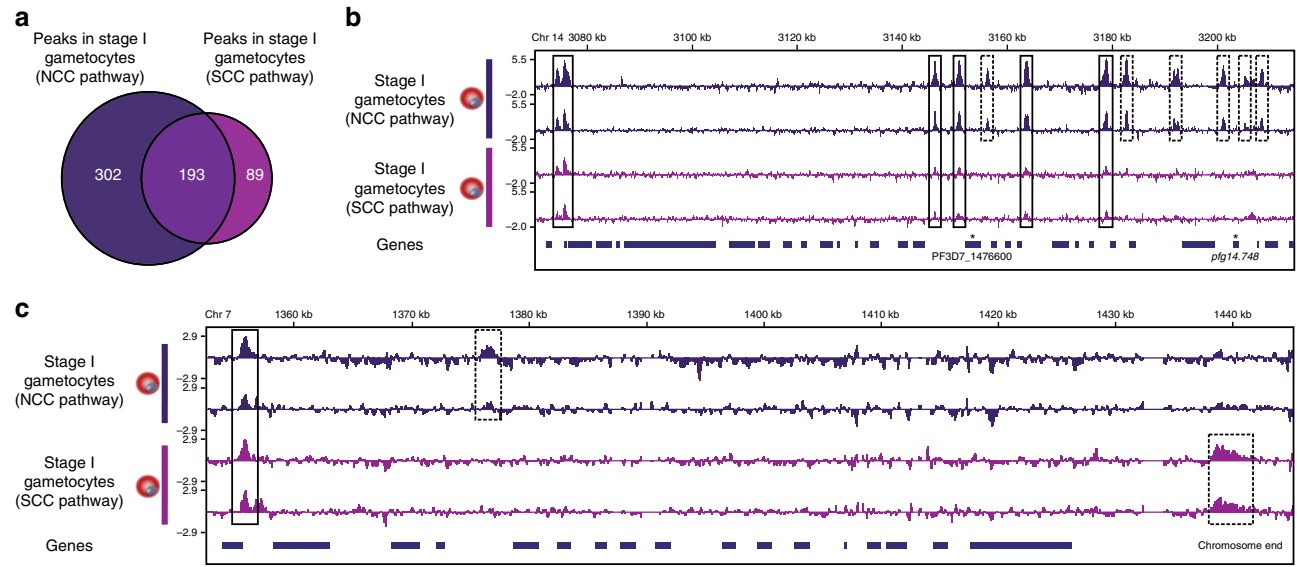

**Fig. 2 Stage I gametocytes resulting from next cycle conversion (NCC) and same cycle conversion (SCC) have major differences in PfAP2-G occupancy. a** Venn diagram showing that although PfAP2-G has many common targets in stage I gametocytes produced via next-cycle conversion and same-cycle conversion, there are also many differences. **b** and **c**, Log2-transformed PfAP2-G ChIP/input ratio tracks for stage I gametocytes derived via NCC (purple) and those derived via SCC (pink) across a region of chromosome 14 (**b**) and the end of chromosome 7 (**c**). The blue bars underneath show the locations of genes. Data are shown for each of two biological replicates per stage. Dashed boxes highlight stage-specific binding sites and solid boxes indicate common binding sites. Asterisks in **b** indicate genes known to be more highly expressed in NCC gametocytes than in SCC gametocytes.

driver of gametocytogenesis. PfAP2-G not only activates genes during the commitment phase but also up until at least stage I of gametocytogenesis. The identification of hundreds of PfAP2-G target genes provides insight into which genes are involved in the early stages of gametocytogenesis.

**PfAP2-G occupancy varies depending on gametocyte conversion pathways.** To further interrogate the function of PfAP2-G in stage I gametocytes, we performed additional ChIP-seq experiments to identify possible differences in PfAP2-G occupancy in gametocytes derived using either the NCC or the SCC route (Supplementary Fig. 1c, d). Previous work identified only three transcripts that were more abundant in NCC cells than in SCC cells, suggesting that the differences between them are fairly minor[3]. In our experiments we sorbitol treated parasites immediately before harvesting them for ChIP to deplete trophozoites and schizonts and leave a relatively pure population of stage I gametocytes.

We found many differences in PfAP2-G occupancy between stage I gametocytes produced using NCC and SCC (Supplementary Datas 6 and 7). Overall, there were more binding sites in NCC cells than in SCC cells (495 vs. 282), with 193 peaks found in both (Fig. 2a). The NCC ChIP samples clustered with the previous stage I gametocyte samples but the SCC samples were more highly correlated with the schizont and sexual ring ChIP samples (Supplementary Fig. 4a). The enriched motifs in regions bound by PfAP2-G in NCC and SCC gametocytes were similar and both resemble the motif found in stage I gametocytes in the previous set of ChIP experiments (Supplementary Fig. 4b, c). The NCC motif is almost identical to the original stage I gametocyte motif whereas the SCC motif lacks a 3′ adenosine and varies in the first two 5′ positions upstream of the GTAC core motif.

Of the three genes previously identified as likely to be NCC specific[3], either no or reduced PfAP2-G binding was observed for two of the three (*pfg14.748* and PF3D7_1476600) in SCC cells (Fig. 2b), whereas the upstream region of the third (PF3D7_0114000) was not significantly bound by PfAP2-G in

either NCC or SCC cells. Beyond this, we found over 200 additional sites bound by PfAP2-G only in NCC cells and 70 that were bound only in SCC cells (Supplementary Data 7). As expected, most of the PfAP2-G binding sites identified in NCC cells and SCC cells were identified in stage I gametocytes in the previous set of experiments (Supplementary Data 7, Supplementary Fig. 4d). PfAP2-G was associated with its own promoter in both NCC and SCC cells. However, SCC-specific peaks were more frequently found only in schizonts and sexual rings, whereas NCC-specific peaks were predominantly found only in stage I gametocytes. Gene Ontology analysis revealed that the genes associated with SCC-specific peaks were weakly enriched in functions related to invasion (represented by only four genes) and genes associated with NCC-specific peaks were most enriched in functions related to glucose catabolism (though this involved only three genes) (Supplementary Data 7). While some binding to invasion gene promoters is seen for both NCC and SCC cells, fewer invasion gene promoters were bound than in schizonts and sexual rings. Most of the invasion genes bound in SCC cells are also bound in NCC cells, including *msrp1*.

The discrepancy between the number of NCC-specific PfAP2-G targets identified here by ChIP-seq but not by Bancells et al. using scRNA-seq could be due to difficulties in detecting low abundance transcripts in scRNA-seq. On the other hand, it is possible that these genes are still transcribed in SCC cells but by a different transcription factor. Interestingly, there were also many SCC-specific binding sites. Many of these were at chromosome ends (Fig. 2c), with 10/14 chromosomes containing PfAP2-G-binding sites at one chromosome end (and sometimes both ends). These sites of occupancy do not occur close to annotated genes so it is difficult to speculate as to what, if any, regulatory function they may have. Interestingly, previous work has shown that gametocytes have an expansion of heterochromatin at the chromosome ends compared to trophozoites[26], so PfAP2-G may play a role in regulating this epigenetic change. These results point to a more restricted function of PfAP2-G in SCC than in NCC and may reflect significant biological differences between these two populations. Future work will be necessary to confirm

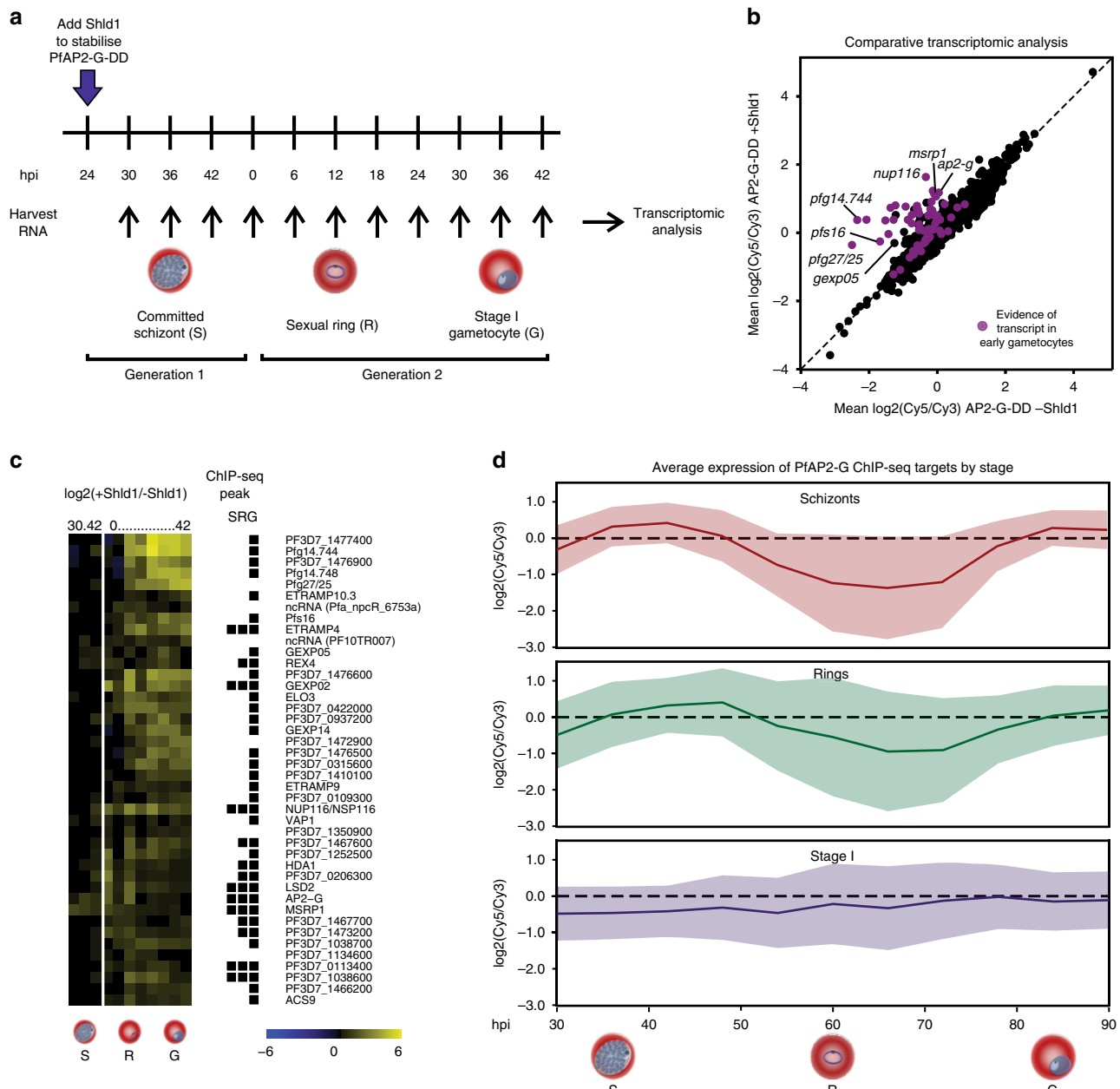

**Fig. 3 PfAP2-G directly regulates many early gametocyte genes. a** Schematic showing the design of the transcriptomic timecourse using AP2-G-DD. The AP2-G-DD line8, which allows for conditional stabilisation or degradation of PfAP2-G depending on the presence of Shld1, was used to identify PfAP2-G target genes. Transcript levels were compared in parallel cultures grown in the presence and absence of Shld1. The timecourse included committed schizonts (S), sexual rings (R), and stage I gametocytes (G). **b** Scatterplot showing mean expression of all transcripts in AP2-G-DD +Shld1 versus AP2-G-DD -Shld1. Transcripts that have been identified in committed schizonts and early gametocytes[13,18,20,27] are in purple. Several genes of particular interest are highlighted. **c** Heatmap showing expression of significantly differentially expressed transcripts (FDR ≤ 0.1 using Significance Analysis of Microarrays, fold change ≥ 1.5) over time in AP2-G-DD +Shld1 compared to AP2-G-DD -Shld1. The black squares to the right indicate whether or not the gene has a ChIP-seq peak upstream of it in each of the three stages tested. **d** Plot showing mean temporal expression of schizont, ring, and stage I gametocyte ChIP-seq targets in the AP2-G-DD line +Shld1. The shaded area indicates the standard deviation. Differences in expression of the three sets of target genes were tested by two-way ANOVA (schizonts vs. rings: p = 0.0097, schizonts vs. stage I: p < 0.0001, rings vs. stage I: p < 0.0001). Source data are provided as a Source Data file.

the extent of these differences and investigate biological consequences arising from differences in PfAP2-G function in NCC and SCC stage I gametocytes.

**PfAP2-G directly regulates the expression of early gametocyte genes**. To supplement our ChIP-seq data, we also used the AP2-G-DD parasite[8] line to examine global transcriptional changes associated with PfAP2-G stabilization. For this timecourse we added Shld1 to AP2-G-DD parasites at the early trophozoite stage (24 hpi) and then collected RNA at six-hourly intervals for 66 h for genome-wide transcriptomic analysis (Fig. 3a). This time period covers the development of committed schizonts into sexual rings and up to stage I gametocytes. As in the case of the ChIP-seq experiments using the AP2-G-DD parasite line, only a subpopulation of cells (~20%) are expected to commit in each cycle in Shld1-treated cultures. We identified 42 genes that were differentially expressed as compared to a parallel culture grown in the absence of Shld1 (false-discovery rate (FDR) ≤ 0.1, fold change ≥ 1.5) (Supplementary Data 8). Consistent with our hypothesis that PfAP2-G is a transcriptional activator and with previous data[20], all differentially regulated genes were upregulated in the presence of PfAP2-G (Fig. 3b). Virtually all of these transcripts are known to be expressed in committed schizonts or early gametocytes[13,18,20,27], including many well-established early gametocyte genes (such as pfs16, pfg14.744, and pfg27) as well as others that have only recently been implicated as potential early gametocyte genes. Of note, only a small number of genes were highly induced in committed schizonts (including ap2-g itself and msrp1), whereas many more genes showed altered expression following reinvasion. Consistent with recent reports that gexp02 is expressed in sexual rings and is thus an early marker of commitment[28], gexp02 is one of the earliest genes to be upregulated following reinvasion.

Importantly, almost all of the genes identified as possible PfAP2-G targets through the expression timecourse were also associated with PfAP2-G binding in at least one stage (Fig. 3c). This includes ap2-g itself, providing evidence for an autoregulatory function as previously suggested[8,9]. Transcript abundance profiles of genes that were associated with PfAP2-G binding in each stage differed significantly from each other (Fig. 3d), with the highest average expression occurring at 42 hpi for schizont ChIP-seq targets, 48 hpi for sexual ring targets, and 78 hpi for stage I gametocyte targets. This suggests that PfAP2-G binding at the promoter of a gene generally co-occurs with an increase in transcript levels for that gene (Supplementary Fig. 5a, b).

A number of potential downstream transcriptional regulators were identified in the transcriptomic analysis, such as the histone deacetylase Hda1 (PF3D7_1472200), the recently identified putative histone demethylase LSD2 (PF3D7_0801900)[20] and another ApiAP2 (PF3D7_1350900 or AP2-O4 in P. berghei)[10]. We also identified genes encoding exported proteins (e.g., gexp02, gexp05, pfg14.744, and pfg14.748) and several encoding proteins involved in fatty acid and lipid metabolism (e.g., elo3 and acs9) using both approaches (Fig. 3c). These findings fit with the known importance of host cell remodeling and lipid metabolism in gametocyte development[29,30].

Notably, gexp05 (PF3D7_0936600) was identified as a target of PfAP2-G in both the ChIP-seq and transcriptomic analyses. GEXP05 is an early gametocyte marker and has previously been shown to be expressed even in a parasite line that does not express functional PfAP2-G and so is thought to be regulated independently of it[31]. Our data suggest that although PfAP2-G is not strictly required for expression of gexp05, its presence nonetheless leads to increased transcription of the gene. This indicates that some degree of transcriptional regulation of

gametocyte genes may occur upstream of PfAP2-G, possibly in a precommitment step that is enhanced once PfAP2-G is present. In line with this, a recent study showed that many gametocyte genes are transcribed even in the absence of functional PfAP2-G but these transcripts are not stabilized, resulting in aborted gametocyte commitment[32].

Although our ChIP-seq data show that PfAP2-G is associated with the promoters of many invasion genes, the only one that is strongly upregulated in our transcriptomic analysis in the presence of PfAP2-G is msrp1 (Supplementary Fig. 6a). MSRP1 has homology to MSP7, a known invasion protein, but little is known about it beyond its dispensability for asexual growth[33]. Multiple datasets show that msrp1 is upregulated in committed cells[17,19,20,34–36]. Crucially, msrp1 is—like ap2-g—highly upregulated in field isolates compared to laboratory strains, suggestive of a possible role in gametocytogenesis[37]. We tested a Δmsrp1 line[33] for its ability to form gametocytes to determine whether MSRP1 plays a role in gametocytogenesis or in invasion by committed merozoites. We found that this line is able to produce gametocytes and thus that MSRP1 is not required for sexual conversion under the conditions tested (Supplementary Fig. 6b). Unexpectedly, the Δmsrp1 line has a significantly higher commitment rate than its parent, which could mean that MSRP1 has a function in sexually-committed cells (though it may also reflect clonal variation between parasite lines).

Overall, invasion genes associated with PfAP2-G binding to their promoters have slightly decreased levels of the corresponding mRNA transcript almost 48 h after peak PfAP2-G binding occurs in schizonts (Supplementary Fig. 7), presumably reflecting the development of a subpopulation of cells (stage I gametocytes) that do not express invasion genes. The delay between peak binding of PfAP2-G and the down-regulation of invasion genes likely rules out a role for PfAP2-G as a direct repressor of invasion genes, though further studies will be necessary to determine the function of PfAP2-G occupancy at these promoters. In agreement with this, the majority of the invasion genes bound by PfAP2-G have recently been shown to be upregulated in committed schizonts using scRNA-seq (Supplementary Data 9)[20,36], suggesting that PfAP2-G may enhance transcription of invasion genes.

**PfAP2-G is an activator of itself and many early gametocyte genes**. Given the potential autoregulation of pfap2-g expression, we sought to demonstrate that binding of PfAP2-G to a promoter leads to an increase in transcript levels by using CRISPR/Cas9 to mutate several putative binding sites upstream of the ap2-g gene. The ap2-g promoter contains eight copies of the PfAP2-G motif spread over 1.3 kb (Fig. 4a). Although the resolution of ChIP-seq is not sufficient to determine which motifs PfAP2-G is binding, based on the position of the PfAP2-G peaks it is possible that all eight are bound. Work in both P. berghei and P. falciparum suggests that AP2-G is regulated in part through a positive-feedback loop in which it is able to bind its own promoter and further increase the levels of ap2-g transcript[8,9,20]. Our ChIP-seq data demonstrate that PfAP2-G directly binds its own promoter, but the functional consequences of this binding are unclear. Therefore, we chose to mutate three (of the eight) motifs that are within 45 bp of each other (Fig. 4a, box) and are positioned in the middle of one of the ChIP-seq peaks. Following successful mutation of the motifs by CRISPR/Cas9 (Supplementary Fig. 8), we found significantly reduced binding of PfAP2-G to its own promoter using ChIP-qPCR (Fig. 4b). We also observed reduced PfAP2-G binding to another region in the ap2-g promoter that was not mutated and to the promoter of a second PfAP2-G target gene (nup116/PF3D7_1473700) (though these differences were not statistically significant).

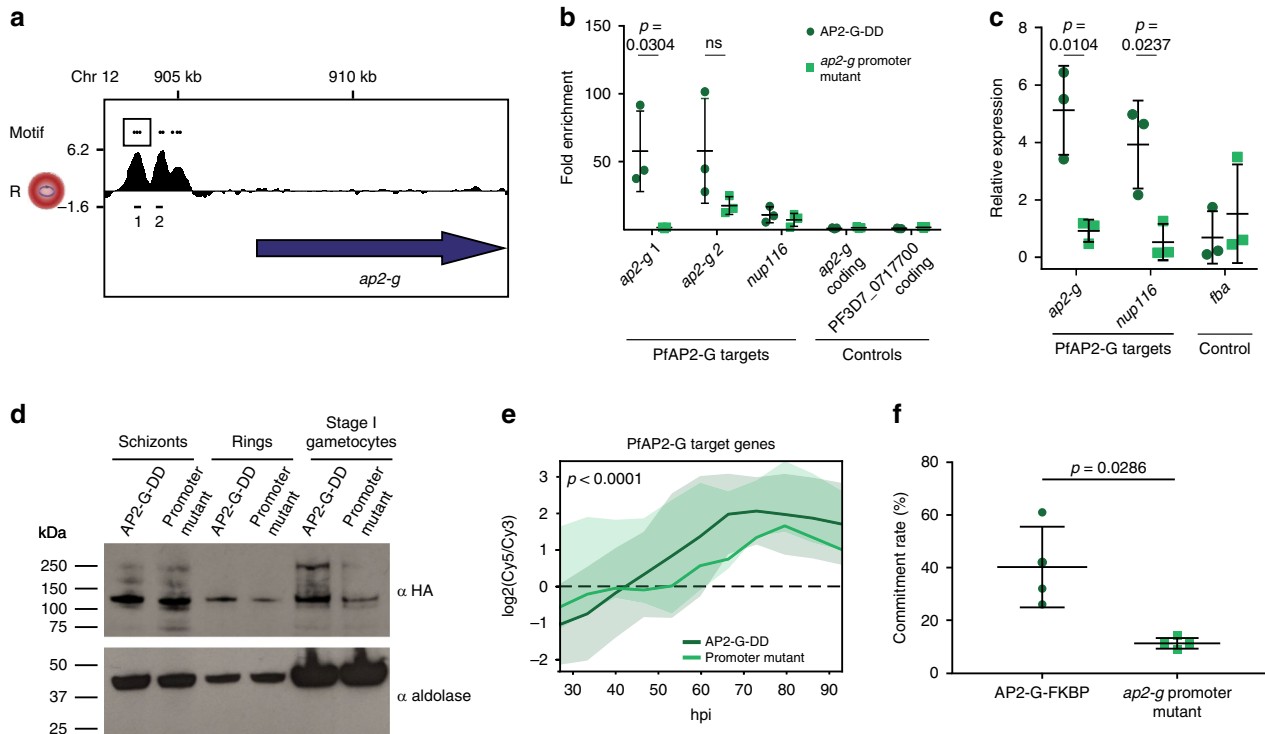

**Fig. 4 PfAP2-G regulates its own transcription. a** Log2-transformed PfAP2-G ChIP/input ratio tracks for the ap2-g locus in sexual rings. The blue arrow shows the coding sequence. The positions of the PfAP2-G motifs (GTACNC) in the promoter are shown above and the box highlights the three that were mutated. The black lines below indicate the regions of the promoter tested by qPCR in panel **c** (not to scale). Data are shown for one of the two biological replicates. **b** ChIP-qPCR performed in ring-stage parasites shows that in the ap2-gpromoter mutant, PfAP2-G is no longer able to bind the region of the promoter containing the mutations. It is still able to bind two other target sites, though at a reduced level compared to its parent. The black bars indicate the mean and standard deviation. n = 3 biologically independent samples. Statistical significance was tested for using t-tests.ns = not statistically significant. **c** qRT-PCR performed in ring-stage parasites shows that mutation of the ap2-gpromoter leads to reduced levels of ap2-gtranscript and that of a target gene. The black bars indicate the mean and standard deviation. n = 3 biologically independent samples. Statistical significance was tested for using t-tests. **d** Western blot showing PfAP2-G protein levels in the AP2-G-DD line and the ap2-g promoter mutant parasite line. Aldolase (bottom panel) was used as a loading control. **e** Plot showing mean temporal expression of the 42 known PfAP2-G targets (from Fig. 1d) in the ap2-g promoter mutant (light green) and the parental line (dark green). The shaded regions show the standard deviation. Differences in expression between the promoter mutant and its parent were tested by two-way ANOVA. **f** The ap2-gpromoter mutant has a lower commitment rate than its parent. The horizontal bars indicate the mean and standard deviation. n = 4 biologically independent samples. The p-value was calculated using the two-tailed Mann-Whitney U test. Source data for panels **b-f** are provided as a Source Data file.

As expected, quantitative real-time polymerase chain reaction (qRT-PCR) showed that mutation of the *ap2-g* promoter also led to significantly reduced levels of *ap2-g* transcript and a target gene (*nup116*/PF3D7_1473700) in rings (Fig. 4c). This reduced *ap2-g* transcript level is reflected in lower abundance of PfAP2-G protein (Fig. 4d), which explains why there is reduced binding of PfAP2-G to sites that were not mutated. As in previous studies[3,8], we see that PfAP2-G is detected as both a full-length protein (284 kDa) and a shorter form (~150 kDa), and that the proportions of the two forms vary by stage. To identify additional changes in transcript levels, genome-wide changes in mRNA abundance were measured. Overall, the *pfap2-g* promoter mutant did not display global changes in transcript levels or altered progression throughout the asexual cycle (Supplementary Fig. 9). Although *ap2-g* levels were not significantly reduced at all timepoints, they were down in the ring stage and overall the expression over time had a reduced range supporting a perturbation of *ap2-g* regulation (Supplementary Fig. 10). qRT-PCR performed at multiple stages confirmed that the downregulation of *ap2-g* in the promoter mutant is restricted to early ring-stage parasites, highlighting this as the period when the autoregulatory function of PfAP2-G occurs (Supplementary Fig. 10d). This reduction in

*ap2-g* levels is evidently sufficient to interfere with PfAP2-G function, as PfAP2-G target genes identified in our initial transcriptional analysis (Fig. 3c) were significantly down-regulated (on average by over 20%) in the *ap2-g* promoter mutant (Fig. 4e, Supplementary Fig. 10, Supplementary Data 10). The largest changes occurred late in the timecourse when most PfAP2-G target genes are usually highly expressed, with 19 of the 42 genes significantly down-regulated across the final four timepoints (FDR ≤ 0.1, fold change ≤ 0.67). As expected, this downregulation of PfAP2-G target genes results in a decrease in sexual commitment rate (Fig. 4f). Overall, these data establish that PfAP2-G indeed regulates itself through a positive feedback loop and confirm its functional role as a transcriptional activator of early gametocyte genes.

**PfAP2-G interacts with the transcription factor PfAP2-I.** We noted that many of the genes whose promoters are bound by PfAP2-G are also targets of the transcription factor PfAP2-I[21]. PfAP2-I is another member of the ApiAP2 DNA-binding protein family that has recently been characterized as an activator of invasion genes[21]. Although ChIP-seq has been previously

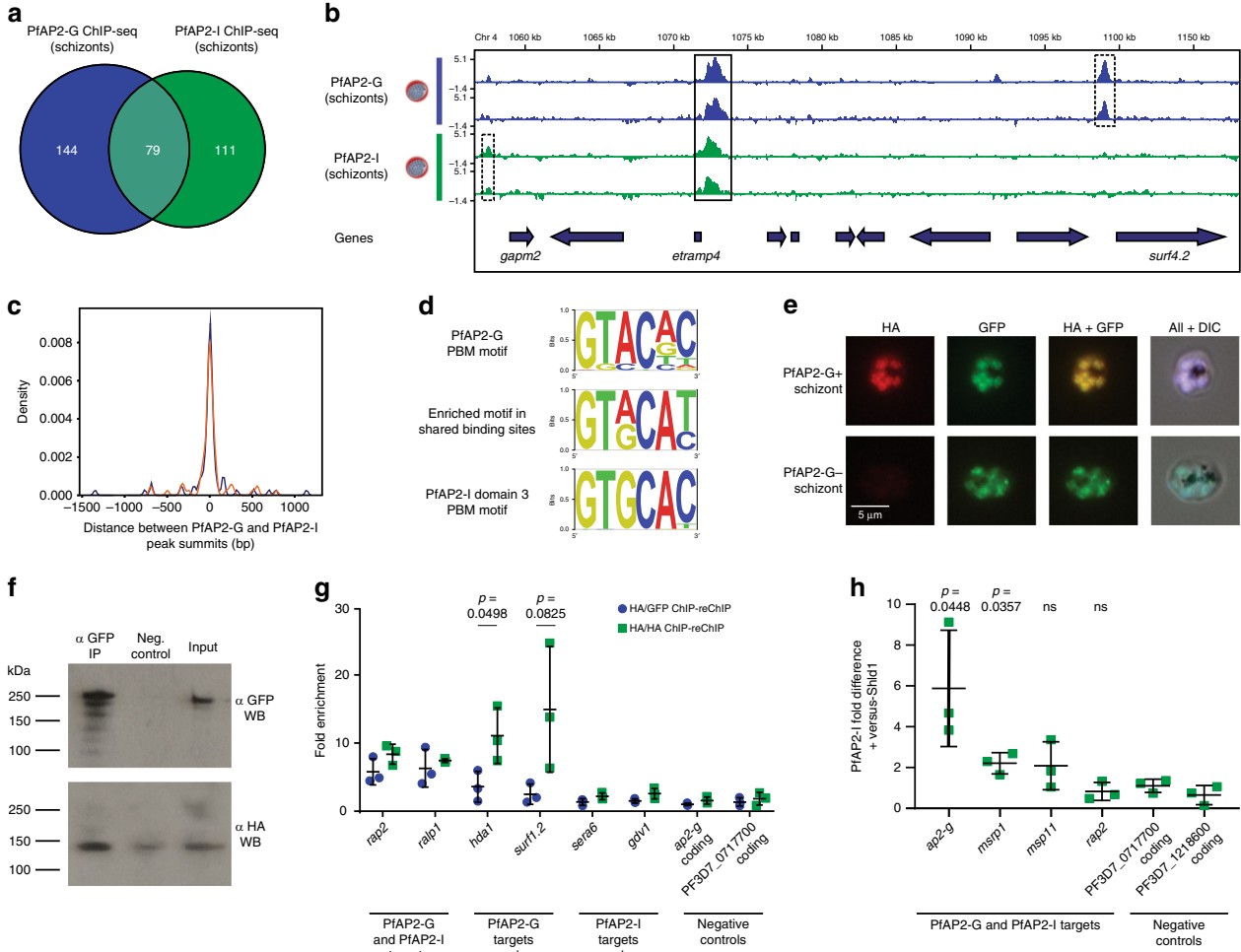

**Fig. 5 PfAP2-G interacts with the transcription factor PfAP2-I. a** Venn diagram showing overlap between PfAP2-G and PfAP2-I binding sites in schizonts in ChIP-seq experiments performed using a parasite line expressing both AP2-G-HA-DD and AP2-I-GFP. **b** Log2-transformed PfAP2-G (blue) and PfAP2-I (green) ChIP/input ratio tracks for schizonts over a region of chromosome 4. Dashed boxes highlight factor-specific binding sites and solid boxes indicate common binding sites. The blue arrows underneath show the locations of genes. Data are shown for each of two biological replicates per factor. **c** Plot showing the distances between PfAP2-G and PfAP2-I peak summits within the 79 regions bound by both proteins. Replicate 1 is shown in blue and replicate 2 is shown in orange. **d** The top enriched motif (determined by DREME) found in the regions bound by both PfAP2-G and PfAP2-I is a composite of the PfAP2-G and PfAP2-I protein-binding microarray DNA motifs. **e** IFA images show that AP2-G+ cells also express PfAP2-I, though not all cells express PfAP2-G. **f** Immunoprecipitation of AP2-I-GFP followed by Western blot with anti-HA and anti-GFP shows that PfAP2-I and PfAP2-G interact. The uncropped original image isprovided as a Source Data file. **g** ChIP-reChIP-qPCR shows PfAP2-G and PfAP2-I bind some gene promoters together in schizonts. Data are represented as fold enrichment relative to a negative control ChIP-reChIP with non-immune IgG. The horizontal bars indicate the mean and standard deviation. n = 3 biologically independent samples. Statistical significance was tested for using t-tests. **h** ChIP performed on PfAP2-I in schizonts shows that some sites have greater enrichment of PfAP2-I in the presence of PfAP2-G, indicative of cooperative binding. Data are represented as the fold change in enrichment of PfAP2-I in a cells treated with Shld1 versus those without Shld1. The horizontal bars indicate the mean and standard deviation. n = 3 biologically independent samples. t-tests were performed to compare the fold change for a region to that of the negative control (Pf3D7_071700 coding). ns = not statistically significant. Raw fold enrichment values are shown in Supplementary Fig. 12. Source data for panels **g**-**h** are provided as a Source Data file.

performed on PfAP2-I, comparing the PfAP2-G and PfAP2-I ChIP-seq datasets is difficult because these experiments were done in parasite lines with different genetic backgrounds (3D7 and Dd2, respectively). For this reason, we endogenously tagged PfAP2-I with GFP in the AP2-G-DD parasite line (Supplementary Fig. 11) and performed ChIP-seq on both proteins in parallel from the same parasite extracts. We confirmed that the two proteins have many shared targets in committed schizonts (Fig. 5a, Supplementary Datas 11 and 12). Although many of the genes associated with binding of both proteins are invasion genes, there are also many that do not have roles in invasion (including three genes encoding

ApiAP2 proteins). Both proteins also have many unique targets (Fig. 5b, Supplementary Data 12); for example, the promoters of *sera6* and *sera7* are bound only by PfAP2-I (though PfAP2-G also binds *sera4*), whereas most early gametocyte genes (such as *pfg27* and *hda1*) are only bound by PfAP2-G. PfAP2-I generally does not bind promoters of genes encoding micronemal or rhoptry neck proteins (as previously described[21]), but PfAP2-G does bind several of these (such as *eba175* and *ron5*). Interestingly, a recent study finds that these genes have increased transcript abundance up until stage I gametocytes, while most other genes involved in invasion are decreased in transcript abundance throughout gametocyte

development[38]. Overall, the summits of the ChIP-seq peaks for the two proteins are extremely close to each other, indicating that they bind the same genomic locations (Fig. 5c). This suggests potential interplay between PfAP2-G and PfAP2-I.

When we performed a motif analysis of regions of the genome bound by both PfAP2-G and PfAP2-I, the top result was a motif which is a composite of the PfAP2-G and PfAP2-I protein-binding microarray (PBM) motifs[22] (Fig. 5d). The core PfAP2-G and PfAP2-I PBM motifs are very similar: the PfAP2-G motif is GTACNC and the PfAP2-I motif is GTGCAC. The common motif had either a G or an A in the third position, giving a motif of GTRCAY. A very similar motif was also identified in a recent study as being highly enriched in a cluster of genes (that includes many invasion genes) with high transcript levels up until stage I of gametocytogenesis and then again in stage V[38]. This hybrid motif suggests that PfAP2-G is not simply recruited by PfAP2-I and tethered to these regions indirectly, but is rather able to associate with at least some of these regions by directly binding DNA. Alternatively, interaction between the proteins may allow a minor change in the DNA-binding preference of either PfAP2-G or PfAP2-I (or both).

To further investigate the potential interaction between PfAP2-G and PfAP2-I, we performed immunofluorescence assays. We found that cells expressing PfAP2-G also express PfAP2-I (Fig. 5e), although PfAP2-G is not present in every cell as has been previously demonstrated[8]. Immunoprecipitation experiments and ChIP-reChIP showed that PfAP2-G and PfAP2-I can directly interact at invasion gene promoters (Fig. 5f, g). Interestingly, two of four tested binding sites (the *ap2-g* and *msrp1* promoters) were more highly bound by PfAP2-I in the presence of PfAP2-G, indicating that the interaction between the two factors is cooperative (Fig. 5h, Supplementary Fig. 12). These findings suggest that PfAP2-G and PfAP2-I cooperatively bind promoters (including those of many invasion genes) in committed schizonts to regulate transcription during sexual commitment.

## Discussion

The master regulator AP2-G and the transcriptional program it directs are essential for sexual differentiation in the malaria parasite[8,9]. We have now identified direct targets of the *P. falciparum* AP2-G transcription factor, and thus the earliest genes expressed during commitment, providing insight into the events occurring during early gametocyte development. Our results overall show that PfAP2-G positively regulates transcription of gametocyte genes, as has been previously hypothesized[8,9]. This is achieved in part by the regulation of PfAP2-G through a positive-feedback loop in which it binds its own promoter to activate transcription, thereby generating high levels of PfAP2-G once the locus initially is activated. Therefore, PfAP2-G establishes a transcriptional program that allows cells to irreversibly differentiate into the sexual stage of the parasite that is critical to mosquito transmission.

The identification of direct targets of PfAP2-G across multiple stages of *P. falciparum* development builds upon and significantly extends findings from previous studies that identified possible PfAP2-G targets using transcriptomic approaches[8,12,20]. Firstly, we demonstrate that PfAP2-G plays a direct role in orchestrating transcription not just during commitment but also into stage I of gametocytogenesis. Secondly, we identify differences in PfAP2-G occupancy between stages (including between stage I gametocytes produced using the NCC and SCC routes) that point to different roles for PfAP2-G over the course of commitment and gametocyte development. Thirdly, we show that many of the possible targets identified in earlier studies are likely not direct targets of PfAP2-G but may be regulated downstream of AP2-G by other effectors. Finally, we identify many additional PfAP2-G targets that previous studies have not linked to PfAP2-G, including genes involved in red blood cell invasion.

As PfAP2-G is a known activator of gametocyte genes and two recent scRNA-seq studies have shown that many invasion genes are upregulated in committed schizonts[20,36], it seems likely that PfAP2-G also activates invasion genes in committed schizonts. Our data further show that PfAP2-G and PfAP2-I interact at many invasion gene promoters, suggesting this interaction could be involved in regulating invasion gene expression in the committed schizont. Notably, PfAP2-I interacts with several proteins with predicted roles in chromatin remodeling[21], so cooperative binding between PfAP2-G and PfAP2-I may allow for remodeling and activation of PfAP2-G target genes. Interestingly, two studies have detected PfAP2-I in gametocytes, indicating that it may play a role beyond the asexual blood stage[30,39].

Combinatorial binding of transcription factors to the same promoters is an important mechanism of gene regulation that both increases specificity and allows finely tuned regulation using a limited pool of transcription factors[40]. Many transcription factors are able to interact in such a way that increases their affinity for DNA, as in the case of enhanceosomes[41]. This may be the case for PfAP2-G and PfAP2-I, as AP2 domains have been shown to dimerize in vitro[42] and cooperative binding of ApiAP2 proteins has been described in the related parasite *Toxoplasma gondii*[43]. An increase in DNA-binding affinity could explain why PfAP2-G binding leads to an increase in transcription of invasion genes. Apart from affecting affinity for DNA, the dimerization of transcription factors can lead to diverse outcomes, including a switch in function from repressing to activating (as in the case of Myc-Max and Mad-Max)[44] and can even change the preferred DNA motif bound by a transcription factor in unpredictable ways[45]. This last possibility is particularly intriguing, as the PfAP2-G ChIP-seq motif only differs from its in vitro motif in the schizont stage, which is when PfAP2-I is also present (Fig. 1c). In future work, the identification of additional proteins that interact with PfAP2-G beyond PfAP2-I will be essential to clarifying how it regulates transcription. One possible interaction partner is PfBDP1, a bromodomain protein that is involved in activation of invasion genes and has been shown to interact with PfAP2-I[21,46].

Our data suggest a model in which PfAP2-G and PfAP2-I cooperate to direct a committed-schizont specific transcriptional program that includes upregulation of many invasion genes. In a committed schizont, many invasion transcripts are slightly upregulated, with *msrp1* alone highly upregulated during commitment across multiple datasets[13,17,19,20,34–36]. One reason for this distinct invasion-related transcriptional program may be to allow committed merozoites to utilize a different invasion pathway than asexual merozoites. This could be analogous to the switch between sialic acid-dependent and -independent invasion pathways, which relies on altered expression of a very small number of genes[47]. A commitment-specific pathway may involve a switch in host cell tropism or reliance on different ligands to enter the red blood cell, or simply an improvement in invasion efficiency that increases the probability that committed cells will survive to form gametocytes and be transmitted. In support of the first possibility, recent data shows that a subset of *P. berghei* merozoites "home" to the bone marrow where they develop within reticulocytes[48]. Importantly, this homing is receptor-mediated, indicating that there may be a transcriptional switch that allows this to occur. De Niz et al. similarly propose that committed merozoites may have a preference for invasion of cells in the bone marrow, and our work provides a possible transcriptional regulatory mechanism for this. Further work will be necessary to establish whether this model is correct or not and to

establish the functional role of PfAP2-G at invasion gene promoters.

In summary, our work shows definitively that the transcription factor PfAP2-G is a major cell fate determinant that triggers sexual commitment of the malaria parasite by activating genes required for early gametocytogenesis and that it continues to play a role beyond the commitment phase. Excitingly, the identification of hundreds of PfAP2-G target genes sheds light on critical processes during commitment. This work significantly advances our knowledge of gametocyte biology and may ultimately help in developing treatments that will prevent this crucial differentiation process and disrupt transmission of the malaria parasite.

## Methods

**Parasites lines.** Parasite cultures were maintained in the presence of 5% $O_2$ and 7% $CO_2$ at 37 °C and grown in RPMI 1640 media supplemented with hypoxanthine and 0.5% Albumax II (ThermoFisher Scientific). Synchronization was performed using 5% sorbitol[49]. Parasites lines used were 3D7, AP2-G-DD[8], AP2-G-DD[ap2-g mut], AP2-G-DD::AP2-I-GFP, and 3D7Δmsrp1[33]. All AP2-G-DD-based lines were maintained in the presence of 2.5 nM WR99210 and routinely cultured in the absence of Shld1. 0.5 μM Shld1 was added only when commitment to gametocytogenesis was required. Shld1 was added at 24 hpi for next-cycle conversion and 6 hpi for same-cycle conversion. The 3D7Δmsrp1 line was maintained in the presence of 2.5 nM WR9910 except when commitment assays were performed. The AP2-G-DD[ap2-g mut] line was created by transfecting uninfected red blood cells simultaneously with pUF-Cas9[50] and pDC2-AP2-Gmut-BSD (described below) and adding AP2-G-DD trophozoites, as described[51]. Starting the following day, cultures were fed using media containing 2.5 nM WR99210, 1 μg/mL blasticidin, and 1.5 μM DSM1. To confirm successful mutation of the promoter, genomic DNA was purified using the Qiagen DNeasy Blood & Tissue kit and the region of interest was PCR amplified and Sanger sequenced. All primers used are listed in Supplementary Data 13. Parasites were then cultured only in the presence of WR99210 and cloned by limiting dilution. The AP2-G-DD::AP2-I-GFP line was made by transfecting the AP2-G-DD line with pSLI-AP2-I-GFP-BSD (described below) and selecting with 2.5 nM WR99210 and 1 μg/mL blasticidin. Once transfectants were obtained, they were selected for integration events using 400 μg/mL G418 as described[52], genotyped by PCR, and then cloned by limiting dilution.

**Measurement of commitment rate.** For AP2-G-DD-based lines, commitment was induced by addition of 0.5 μM Shld1 at the trophozoite stage (24 hpi). 3D7Δmsrp1 and its parental line were grown for at least two weeks in the presence of 200 μM choline to suppress commitment[34] and choline was removed in the trophozoite stage in the cycle the commitment assay began. For these experiments 3D7Δmsrp1 was grown in the absence of WR99210. Cultures were treated with 50 mM N-acetyl glucosamine to kill asexual parasites for the four days following addition of Shld1 or the removal of choline, with daily media changes. The commitment rate was calculated as the gametocytemia 7 days after addition of Shld1 or removal of choline divided by the ring parasitemia the day following addition of Shld1. Four biological replicates were performed. Statistical comparisons were performed using the two-tailed Mann–Whitney U test.

**DNA constructs.** All primers used are listed in Supplementary Data 13.

To mutate the promoter of ap2-g using CRISPR/Cas9, complementary oligonucleotides encoding the guide RNA were inserted into pDC2-U6A-hDHFR (kind gift of Marcus Lee) using BbsI sites. A 633 bp homology region containing the three mutated PfAP2-G motifs (GTAC → GATC) was generated using overlap extension PCR, cloned into pGEM-T Easy (Promega), and then inserted using the NotI site. Finally, the hDHFR cassette was replaced with a bsd cassette obtained through PCR from pBcamHA using the NotI and SacI sites. The resulting plasmid was pDC2-AP2-Gmut-BSD.

The plasmid used to tag AP2-I with GFP in the AP2-G-DD line (pSLI-AP2-I-GFP-BSD) was modified from pSLI-TGD[52]. The hDHFR cassette was replaced with the bsd cassette from pBcamHA using the BamHI and HindIII sites and then a PCR product containing the 3′ end of ap2-i was inserted upstream of and in frame with gfp using the NotI and MluI sites.

**Chromatin immunoprecipitation.** For ChIP-experiments, Shld1 was added to AP2-G-DD or AP2-G-DD::AP2-I-GFP cultures at 24 hpi to stabilize PfAP2-G and nuclei were harvested 12-h (committed schizonts), 36-h (sexual rings), or 60-h (stage I gametocytes) later. For stage I gametocytes generated via same-cycle conversion, Shld1 was instead added at 6 hpi. Stage I gametocytes for the NCC and SCC comparison experiments were treated with 5% sorbitol prior to crosslinking and chromatin harvest to kill asexual parasites[53]. Two biological replicates were used for ChIP-seq experiments and three for ChIP-qPCR. Cultures were incubated with 1% formaldehyde for 10 min at 37 °C to crosslink. After quenching with 100 mM glycine, cross-linked parasites were released from red blood cells using 0.1%

saponin in PBS and washes extensively. Parasites were lysed following incubation in lysis buffer (10 mM Hepes (pH 7.9), 10 mM KCl, 0.1 mM EDTA (pH 8.0), 0.1 mM EGTA (pH 8.0)) for 30 min by adding 0.25% Nonidet P-40 and douncing. Cross-linked nuclei were then washed in lysis buffer, resuspended in Covaris shearing buffer (0.1% sodium dodecyl sulfate (SDS), 10 mM Tris pH 8.1, 1 mM EDTA) and sonicated using an M220 ultrasonicator (Covaris) and the following conditions: 5% duty cycle, 75 W peak incident power, 200 cycles per burst, treatment time 300 s. The sonicated chromatin was diluted tenfold with dilution buffer and precleared with Protein A/G magnetic beads (Pierce) for 2 h. The pre-cleared material was then incubated with antibody and magnetic beads overnight, with a small volume of non-immunoprecipitated input material kept separately. For immunoprecipitations with AP2-G-DD, rat α HA (Roche 3F10) was used and for AP2-I-GFP rabbit α GFP (Abcam ab290) was used. For ChIP-qPCR, negative control IPs were also performed using either rat IgG (Abcam ab37361) or rabbit IgG (Abcam ab46540). The next day, beads were washed with low salt buffer (0.1% SDS, 1% Triton X-100, 2 mM EDTA (pH 8.0), 20 mM Tris–HCl (pH 8.0), 150 mM NaCl), high salt buffer (0.1% SDS, 1% Triton X-100, 2 mM EDTA (pH 8.0), 20 mM Tris–HCl (pH 8.0), 500 mM NaCl), LiCl buffer (0.25 M LiCl, 1% Nonidet P-40, 1% sodium deoxycholate, 1 mM EDTA (pH 8.0), 10 mM Tris–HCl (pH 8.0)), and TE buffer. The bound DNA was then eluted in elution buffer (1% SDS, 100 mM NaHCO₃). Cross-linking was reversed by incubating overnight at 45 °C and DNA was subsequently treated with RNase A at 37 °C for 30 min and Proteinase K at 45 °C for 2 h. DNA was purified using the Qiagen MinElute kit and quantified using the Qubit HS DNA assay. ChIP DNA was either used to make libraries for ChIP-seq or analyzed by qPCR as described below.

For ChIP–reChIP, the first ChIP was performed as described above up to the elution step. Beads were incubated with 10 mM DTT in TE for 30 min at 37 °C and the recovered eluted DNA was then diluted 20-fold in dilution buffer. The diluted material was then subject to a second round of immunoprecipitation and eluted and processed as described above. Totally, 0.2 mg/mL HA peptide was included in the immunoprecipitation with α GFP to prevent reprecipitation with α HA still present.

**qPCR.** Primers used for qPCR analysis are listed in Supplementary Data 9. All data were obtained using an Applied Biosystems 7300 real-time PCR machine and SDS v1.4 (Applied Biosystems). Data were analyzed using the ΔΔCt method. All primer sets used for qPCR were first tested using serial dilutions of 3D7 genomic DNA to determine efficiency (primers were only used if >90%) and specificity (based on the presence of a single peak in the melting curve). For qRT-PCR, PF3D7_0717700 was used as a normalizing gene. For ChIP-qPCR, values are presented as fold enrichment in the immunoprecipitated sample vs. the negative control (a mock IP performed with non-immune IgG). For the AP2-I ChIP-qPCR in the AP2-G-DD::AP2-I-GFP line without and with Shld1, values are additionally presented as fold enrichment +Shld1 vs. −Shld1. All qPCR experiments used three biological replicates. Statistical significance was tested for using unpaired t-tests.

**Library construction and analysis.** Libraries for sequencing were constructed using NEBNext DNA library prep reagents (New England Biolabs) according to the manufacturer's instructions (except for the PCR step). Following end repair, purification with Agencourt AMPure XP beads (Beckman Coulter), dA-tailing, and purification, NEXTflex DNA barcodes (Bioo Scientific) were ligated to the DNA fragments. Following another purification, Ampure XP beads were then used to size select for 250 bp inserts. Kapa HiFi DNA Polymerase was used to PCR amplify the libraries, which were then purified and quantified using the Qubit dsDNA HS Assay kit (Life Technologies). Library quality was assessed using the Agilent 2100 Bioanalyzer (Agilent Technologies). Sequencing was performed using an Illumina Hiseq 2500 or Illumina Nextseq 550 to generate 150 bp single-end reads.

**ChIP-seq data analysis.** Following assessment of the quality of the reads using FastQC[54], adapter sequences were trimmed using Trimmomatic[55] and reads mapped to the P. falciparum 3D7 genome (Pf 3D7 v28, obtained from PlasmoDB) using BWA-MEM[56]. Reads were filtered using SAMtools[57] to remove duplicates. For visualization, bam files were converted to bigwig files showing log2(IP/input) using bamcompare from the deeptools suite[58] and viewed using IGV[59]. MACS2[60] was used to call peaks with a q value cutoff of 0.01. Called peaks were annotated using BEDtools[61]. For peaks from individual replicates, distances given are from the peak summit to the nearest gene. For peaks resulting from the overlap of multiple replicates, distances given are from the center of the peak to the nearest gene. Two biological replicates were used for each experiment and subsequent analysis focused on regions bound only in both samples (identified using the Multiple Intersect function of BEDtools[61]). The subtractBed function of BEDtools[61] was used to identify regions that are stage- or factor-specific. Peaks identified as stage- or factor-specific are present in both biological replicates for the first condition but do not overlap either biological replicate for the second condition. The Multiple Intersect function of BEDtools[61] was used to identify peaks found in NCC and SCC stage I gametocytes that overlap with those found in other stages.

DREME[62] was used to identify enriched DNA motifs in sequences associated with called peaks and TOMTOM[63] was then used to compare any identified motifs with known ApiAP2 motifs[22]. Sequence logos were generated use enoLOGOS[64]. Gene ontology analysis was performed using the gene ontology enrichment tool in

PlasmoDB with a p value cutoff of 0.01[65]. The plotProfile tool in the deeptools suite[58] was used to plot the average enrichment of PfAP2-G in the 1000 bp upstream of the 36 invasion genes it is associated with. The correlation heatmap of PfAP2-G ChIP-seq experiments was made using multiBigwigSummary and plotCorrelation in the deeptools suite[58]. The inputs were input-normalized bigwig files produced as described above and only regions that were associated with peaks in at least one stage were considered. Values given are Pearson correlation coefficients.

For the generation of motif heatmaps, instances of the most enriched overall motifs for each dataset were identified within the overlapping peaks using FIMO[66], with a significance threshold of 1e-3 against a second order Markov model. The top scoring motif was retained for each peak. A four color nucleotide plot was generated using the cegr-tools program FourColorPlot, available at https://github.com/seqcode/cegr-tools. Each nucleotide sequence was centered on the top scoring instance of the motif of interest, and a ten nucleotide extension was added in either direction of the conserved motif occurrence.

Genes that were within 2 kb downstream of peaks called in both replicates in stage I gametocytes were cross-compared to known male and female-enriched transcripts and proteins[23]. Differentially expressed proteins and transcripts in either male or female gametocytes were defined as those with at least fourfold more protein/transcript in the respective gametocyte sex. Statistical overrepresentation of male and female proteins and transcripts in PfAP2-G targets were tested with a Fisher's exact test in R.

**RNA purification and cDNA synthesis**. RNA was extracted from synchronized parasites by resuspending infected red blood cells in TRIzol (ThermoFisher Scientific) and following the manufacturer's protocol. For qRT-PCR, cDNA was synthesized from 2 μg RNA using SuperScript II Reverse Transcriptase (Thermo-Fisher) with random nonamers and oligo(dT) primers. Negative controls without reverse transcriptase were also prepared to determine by qPCR whether contaminating gDNA was present.

**DNA microarrays**. RNA was extracted from synchronized PfAP2-G-DD parasites (±Shld1) using TRIzol (ThermoFisher Scientific) and following the manufacturer's protocol. Purified RNA was reverse transcribed into cDNA using Superscript II RT (ThermoFisher) according to the manufacturer's protocol with the inclusion of dUTP-aminoallyl nucleotide as well as a mixture of random nonamer and oligo-dT primers. The resulting aminoallyl cDNA was concentrated using DNA Clean and Concentrator-5 (Zymo Research) according to the manufacturer's protocol. Aminoallyl-cDNA was labeled using Amersham CyDye Reactive Dyes (GE Healthcare) according to the manufacturer's instructions, and purified using DNA Clean and Concentrator-5 kit. Cy5 was used to label each time point sample and Cy3 for the reference pool, which consisted of cDNA from an asexual cDNA pool plus new cDNA from the timecourse samples (to make up 50–63% of the new reference pool) to ensure that gametocyte-specific transcripts would be well represented in the reference and thus could be quantified appropriately. Two-channel Agilent DNA microarrays (AMADID 037237) were hybridized and washed before scanning on an Axon 4200A scanner using established methods[21,67]. After scanning, signal intensities for each gene were extracted from the scanned image using Agilent Feature Extraction Software version 9.5 employing the GE2_1100_Jul11_no_spikein extraction protocol. Detailed protocols can be found at http://llinaslab.psu.edu/protocols/. Data were analyzed using the Significance Analysis of Microarrays[68] package in R. Two-way ANOVA was used to test for differences in transcript abundance profiles between stages. In scatter plots showing microarray data, the Lowess curve is shown in addition to the individual data points.

**Immunofluorescence assays**. Immunofluorescence assays were performed following fixation with 4% formaldehyde and 0.0075% glutaraldehyde in PBS for 30 min at room temperature[69]. After quenching with 0.1 mg/ml NaBH₄ in PBS and blocking with 3% bovine serum albumin in PBS, PfAP2-G and PfAP2-I were detected with 1 μg/mL rat α HA (Roche 3F10) and 2 μg/mL rabbit α GFP (Abcam ab290), respectively. Secondary antibodies used were 2 μg/mL AlexaFluor 546 coupled goat α-rat (ThermoFisher Scientific) and 2 μg/mL AlexaFluor 488 coupled goat α-rabbit (ThermoFisher Scientific). All washes were performed using PBS. Images were obtained using an Olympus BX61 system and SlideBook 5.0 (Intelligent Imaging Innovations).

**Immunoprecipitation**. Immunoprecipitation experiments were performed using 36 hpi AP2-G-DD::AP2-I-GFP schizonts to which Shld1 had been added 12 h earlier. Nuclear proteins were extracted using a conventional protocol[70], and then diluted one-fourth in dilution buffer with 0.025% Tween-20 and added to washed GFP-Trap_MA beads (Chromotek). For the negative control, binding control magnetic agarose beads (Chromotek) were used. Following incubation at 4 °C for 1 h while rotating and five washes with PBS-T, bound proteins were eluted by boiling in loading buffer.

**Western blot**. Proteins were run on a 4–12% Bis–Tris gel (ThermoFisher) in MES buffer and then transferred to a nitrocellulose membrane. After blocking with 5% milk in PBS-T for 30 min, membranes were incubated overnight with the primary antibody in 5% milk. Primary antibodies used were: 1 μg/mL rat α HA (Roche 3F10), 1/2000 rabbit α GFP (Abcam ab290), 1/1000 rabbit α aldolase HRP (Abcam ab38905), or 1/3000 mouse α H3 (Abcam ab10799). The next day, membranes were washed three times with PBS-T and incubated for 2 h with the appropriate secondary antibody, then washed again. Secondary antibodies used were: 1/3000 goat anti-rat HRP conjugate (Millipore), 1/10,000 goat anti-rabbit HRP conjugate (Millipore), or 1/5000 goat anti-mouse HRP conjugate (Pierce). ECL reagent (Pierce) was used to detect bound antibody. The uncropped original images for all Western blots shown are provided as a Source Data file.

**Reporting summary**. Further information on research design is available in the Nature Research Reporting Summary linked to this article.

## Data availability

ChIP-seq data are deposited in the NCBI Sequence Read Archive (SRA) under the accession numbers GSE120448 (PfAP2-G enrichment in committed schizonts, sexual rings, and stage I gametocytes), GSE134268 (PfAP2-G enrichment in NCC and SCC stage I gametocytes), and GSE120488 (PfAP2-G and PfAP2-I enrichment in schizonts). Microarray data are deposited in the NCBI Gene Expression Omnibus (GEO) under the accession numbers GSE120990 (AP2-G-DD + vs. − Shld1) and GSE121312 (AP2-G-DD$^{ap2-g\ mut}$ and AP2-G-DD). The source data underlying Figs. 3d, 4b–f, 5f–h, and Supplementary Figs. 6b, 7, 10d, 11c, d, and 12 are provided as a Source Data file.

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

## Acknowledgements

This work was funded through NIH grant R01 AI125565 and generous support from The Pennsylvania State University. G.A.J. is a recipient of the Sir Keith Murdoch Fellowship from the American Australian Association and a Postdoctoral Research Grant from the American Heart Association (16POST26420067).

## Author contributions

G.A.J., T.J.R., and M.L. designed the experiments. G.A.J., T.J.R., J.V., L.O., and H.P. performed the experiments. G.A.J. and T.J.R. analyzed the data and generated figures. R. V.B. analyzed the data. G.A.J. and M.L. wrote the paper, with input from all the other authors.

## Competing interests

The authors declare no competing interests.
