## [Peer Review File · Nature Communications]

Editorial Note: This manuscript has been previously reviewed at another journal that is not operating a transparent peer review scheme. This document only contains reviewer comments and rebuttal letters for versions considered at Nature Communications .

Reviewers' Comments:

Reviewer #1:

Remarks to the Author:

Reviewer 1 gave an extensive summary of the main findings of the paper. I agree with Reviewer 1's assessment that the interaction between AP2-G and AP2-I is the most interesting part of this manuscript, but also think that the AP2-G ChIP-seq data adds significantly to our understanding of the cascade of regulation that takes place during gametocyte commitment and the following stages. In terms of continuity of the manuscript, it bothered me a bit that the invasion gene story is chopped up into three parts and I wondered if the manuscript can be streamlined to increase readability.

Most of Reviewer 1's comments have been addressed, and the suggestions that are not addressed (such as FRET to show interaction between AP2-G and AP2-I) are in my opinion not strictly necessary given the data already presented in the manuscript (i.e., ChIP-reChIP).

However, I do have several issues with the manuscript that were not raised by Reviewer 1. In multiple cases, data is selectively presented, no tests were performed to determine whether differences are statistically significant, and conclusions are overstated. In particular:

Line 162 – 163: It looks like SCC ChIP-seq signal-to-noise is lower than NCC signal, resulting in possibly missing peaks in the SCC samples that are observed in the NCC samples. I understand that this is probably a result of technical challenges of performing ChIP on a TF that is expressed in a subpopulation of cells. However, the conclusion that SCC and NCC pathways are very different may be preliminary and should be made cautiously. Also, Suppl. Fig 4 shows fairly low correlations between biological replicates and I have to say that in my opinion the color scheme used here is misleading – I would expect to have high correlations depicted in red and low correlations in blue, so on first glance it looks great, but the numbers are actually disappointing.

Lines 196 – 197: The statement "The timing of PfAP2-G binding at the promoter of a gene generally corresponded well with when an increase in levels of the transcript under its control occurred" is not sufficiently substantiated by the two examples in Fig. 3d/e. In Fig. 3c, I can find many examples of genes for which transcription and PfAP2-G ChIP signal do not completely agree. A more general statistical analysis of this correlation is warranted.

Lines 228 – 230: The difference in transcript levels of invasion genes in the absence and presence of Shield is not statistically tested and given the overlapping SD curves in Suppl. Fig 6, I wonder whether this statement is justified.

Supplementary Fig 9: differences presented here and described in the main text should be tested for statistical significance.

Minor comments:

Line 94 should read: including many *that* are stage-specific.

Figure 3: Check panel numbering in legend – ‘e’ and ‘f’ refer to panels ‘d’ and ‘e’. Line 203 refers to panel 3d which I believe is not presented in this version of the manuscript.

Reviewer #3:

Remarks to the Author:

This manuscript by Josling et al. dissects the role of the AP2-G (previously identified master switch of the sexual commitment in the malaria parasites) in the regulation gene expression throughout the gametocyte generation process. The authors used the *Plasmodium falciparum* line in which C-terminally-tagged AP2-G is put under the control of the inducible expression system to synchronously induce gametocytogenesis and analyse the binding profile of AP2-G and the transcriptome at different time-points of this process. They also compared the AP2-G targets in the same cycle vs. next cycle committed parasites and use the cas9-mediated mutagenesis to confirm the previously postulated role of AP2-G indirectly regulating its own expression. Importantly the manuscript shows the unexpected interplay between AP2-G and another transcription factor (AP2-I), with both factors binding overlapping set of promoters at the schizont stage.

This paper represents a significant advancement in the field of gametocytogenesis and gene expression regulation in *Plasmodium*, providing first comprehensive list of the direct AP2-G targets and revealing the role this protein may play in the regulation of the invasion related genes. The main conclusions are clearly stated, backed-up by the data from multiple sources and the manuscript is of high interest to both the malaria researchers and to the parasitology community in general. As I was asked to provide my opinion regarding the comments of the previous reviewer rather than an independent review, I will start by addressing them

1. PfAP2-G regulates the expression of many early gametocyte genes. The PfAP2-G DD line has been used in Poran et al, Nature 2017 and Kafsack et al., Nature 2014. The first paper analyses global transcriptional differences using a single cell approach on synchronized parasites with time points extremely similar to this submitted paper. The second paper shows that PfAP2-G is upregulated in gametocyte producing lines and the subsequent transcriptional profile of knockdown show a cluster of downregulated genes is highly enriched for genes expressed during the first stages of gametocyte formation. In Line 118, the authors state “PfAP2-G is a direct activator of early gametocyte genes and ... the key driver of gametocytogenesis.” My argument would be that the Poran and Kafsack papers show this already.

The causative link the AP2-G expression and gametocytogenesis has been indeed shown before in at least 6 manuscripts (Kafsack et al, 2014, Poran et al, 2017, Sinha et al, 2014, Kent et al., 2018, Bancells et al., 2019, Brancucci et al., 2018). The authors themselves are acknowledging this in the introduction, citing the relevant papers and providing a brief summary of the past findings. However, none of the previous papers could confidently discriminate between the direct targets of AP2-G and other genes, co-induced during the early gametocytogenesis by secondary factors. They were also unable to detect the genes, which, while regulated by AP2-G, were not showing strong changes in expression in the committed population. I would therefore agree with the author statement: “We have now identified direct targets of PfAP2-G and thus the earliest genes expressed during commitment, providing unprecedented insight into the events occurring during early gametocyte development.”. I believe that the current version of the manuscript addresses this point in a satisfactory way.

2. PfAP2-G regulates genes encoding proteins important for red blood cell invasion. While the authors show that PfAP2-G was associated with promoters of many genes in red blood cell invasion, only one gene *msrp1* shows a different expression profile with or without PfAP2-G. Furthermore, they observe that AP2-G and AP2-I bind to the promoters of these invasion genes. While the data looks convincing

that AP2-G and AP2-I may be binding as close or similar sites, the authors extend this observation to make a statement that the regulation of gametocytogenesis and invasion are linked during malaria parasite development. I am not fully convinced that the authors have enough mechanistic data to make that link. One potential experiment is to determine if the knockdown lines have a defect in invading reticulocytes using parasite growth assays and how this changes the dynamics of commitment. Having alternative biological assays looking at invasion could provide a more mechanistic link rather than a correlation of binding sites of particular cluster of genes.

I agree with the reviewer that one needs to be careful regarding the interpretation of the binding of AP2-G to the promoters of invasion genes. The lack of detectable expression changes in most of these genes combined with the fact that AP2-G expression in schizonts is not a pre-requisite for successful gametocytogenesis, suggests that its role at this stage is at best modulatory. While it is certainly tempting to speculate that committed merozoites exhibit different invasion patterns and AP2-G is responsible for it, at the moment there is little data to substantiate these claims. However, I don't believe that additional experiments tackling this question are necessary to complete the manuscript, as they would require a substantial amount of work, which would be out of the scope of the topic of this work – identification of the AP2-G targets. Authors should make clear throughout the manuscript that their interpretation of the role of AP2-G in schizonts is purely speculative and would require further in-depth studies.

3. Motif identification for PfAP2-G and PfAP2-I and mechanism of binding. It would have been useful to see a more thorough examination of the putative motif site and data towards the independent/cooperative binding between these two transcription factors. Using EMSA as the Llinas lab has done before, they could interrogate the critical residues of the motif using mutated sites and also the sequence of binding (whether independent or at the same time).

The current version of the manuscript provides a good overview of the motif bound at different life stages by AP2-G (see Fig S2). I agree that it would be useful to compare the relative binding affinity of the AP2 domains of AP2-G and AP2-I to the consensus sequences enriched within the shared peaks and to each other in vitro motifs. As the binding specificity of both domains have been extensively studied by Llinas group using protein binding microarrays containing all possible variants of the 8bp DNA motifs (Campbell et al., 2010), perhaps the relevant information could be extracted from that dataset?

I also have two additional comments regarding the manuscript itself:

4. One of the more interesting findings of the manuscript is the fact that AP2-G appears to bind different sets of targets depending on the commitment pathway (NCC and SCC). However, besides noting this fact, authors provide little information regarding these differences. It would be helpful to see additional analysis including GO terms enriched in SSC- and NCC- exclusive gene sets, the consensus sequence motifs bound in the two datasets, and comparison with the previous AP2-G invasion profiles (eg. are genes bound exclusively in the NCC pathway the ones that are also bound in committed schizont, ring stages? Is AP2-G binding to its own promoter in both pathways? Is the MSRP1 binding present in SSC pathway even if these parasites won't pass through the merozoite stage?). That would presumably allow to generate some testable hypotheses regarding the differences in the AP2-G function in the two pathways.

5. Gametocytes generated by Plasmodium can be either male or female. While the two populations cannot be morphologically separated till later in the development, it was previously speculated that the sex of gametocyte is predetermined during the commitment as all merozoites from the same

schizonts would all differentiate into males or females (Silvestrini et al, 2000). Stage I gametocyte pool studied by the authors would therefore contain two separate populations. Are the genes induced in the stage I gametocytes by AP2-G expressed in both male and female gametocytes later in the infection, or some of them show strong sex-specific bias? Adding an extra column to one of the supplementary data tables, specifying this information (available from Lasonder et al., 2016) would answer this question and provide some information regarding the role of the AP2-G in sex determination.

Since the reviewers for *Nature Communications* have been asked to review what previous reviewers have commented on in our previous *Nature Microbiology* submission, we have made an effort to address all comments and concerns throughout the manuscript and have made numerous changes which we believe strengthen the manuscript overall. We hope you will agree.

***Nature Communications* Reviewer 1:**

Reviewer 1 gave an extensive summary of the main findings of the paper. I agree with Reviewer 1's assessment that the interaction between AP2-G and AP2-I is the most interesting part of this manuscript, but also think that the AP2-G ChIP-seq data adds significantly to our understanding of the cascade of regulation that takes place during gametocyte commitment and the following stages.

We thank the reviewer for this comment and agree that our work extends the work of others and is the first definitive demonstration of the genes that are directly regulated by PfAP2-G that are not merely changing due to a change in state (from asexual to sexual).

In terms of continuity of the manuscript, it bothered me a bit that the invasion gene story is chopped up into three parts and I wondered if the manuscript can be streamlined to increase readability.

We agree with the reviewer that the original submitted version spreads our observations regarding the regulation of invasion into separate sections. This change was made in response to the original set of reviews to avoid over-emphasising the invasion data. We have endeavoured to integrate our results and discussion based on the impact on the regulation of invasion throughout the manuscript.

Most of Reviewer 1's comments have been addressed, and the suggestions that are not addressed (such as FRET to show interaction between AP2-G and AP2-I) are in my opinion not strictly necessary given the data already presented in the manuscript (i.e., ChIP-reChIP).

We thank the reviewer for realizing that we present alternative data which similar makes the same case in several instances.

However, I do have several issues with the manuscript that were not raised by Reviewer 1. In multiple cases, data is selectively presented, no tests were performed to determine whether differences are statistically significant, and conclusions are overstated. In particular:

Line 162 – 163: It looks like SCC ChIP-seq signal-to-noise is lower than NCC signal, resulting in possibly missing peaks in the SCC samples that are observed in the NCC samples. I understand that this is probably a result of technical challenges of performing ChIP on a TF that is expressed in a subpopulation of cells. However, the conclusion that SCC and NCC pathways are very different may be preliminary and should be made cautiously.

We agree and have attempted to make claims about differences between NCC and SCC cells more cautious. For example, we have added the sentence: "Future work will be necessary to confirm the extent of these differences and investigate further possible differences in PfAP2-G function in NCC and SCC stage I gametocytes." We have also removed 'major' from the original subheading "There are major differences in PfAP2-G occupancy in stage I gametocytes originating from different conversion pathways".

Also, Suppl. Fig 4 shows fairly low correlations between biological replicates and I have to say that in my opinion the color scheme used here is misleading – I would expect to have high correlations depicted in red and low correlations in blue, so on first glance it looks great, but the numbers are actually disappointing.

We apologise that the original colours in Supplementary Fig. 4 were misleading. The palette used was the default for the plotCorrelation tool, but we have now changed it. In the original version of this figure the correlations were low because we included all regions of the genome in the analysis. We have now repeated this analysis restricting the comparison to regions that are associated with peaks in at least one stage. Because PfAP2-G has a fairly restricted distribution, we believe this is a more meaningful comparison than the original. In this new analysis we still see that the NCC samples correlate best with the Stage I gametocyte samples, whereas the SCC samples are more weakly correlated.

Lines 196 – 197: The statement “The timing of PfAP2-G binding at the promoter of a gene generally corresponded well with when an increase in levels of the transcript under its control occurred” is not sufficiently substantiated by the two examples in Fig. 3d/e. In Fig. 3c, I can find many examples of genes for which transcription and PfAP2-G ChIP signal do not completely agree. A more general statistical analysis of this correlation is warranted.

We thank the reviewer for this suggestion. We have now performed a more robust analysis of the relationship between timing of PfAP2-G binding and transcript abundance profiles (Fig 3d) and moved the original Fig 3d, e to Supplementary Fig. 5. The new analysis shows that transcript levels of genes associated with binding in different stages vary significantly over time and the time of peak transcript levels for each group corresponds to the stage at which the ChIP was performed.

Added to results: “Transcript abundance profiles of genes that were associated with PfAP2-G binding in each stage differed significantly from each other (Fig. 3d), with the highest average expression occurring at 42 hpi for schizont ChIP-seq targets, 48 hpi for sexual ring targets, and 78 hpi for stage I gametocyte targets. This suggests that timing of PfAP2-G binding at the promoter of a gene generally corresponds well with when an increase in levels of the transcript under its control occurred (Supplementary Fig. 5a, b).”

Lines 228 – 230: The difference in transcript levels of invasion genes in the absence and presence of Shield is not statistically tested and given the overlapping SD curves in Suppl. Fig 6, I wonder whether this statement is justified.

We apologise for not including a statistical test in the original analysis. We have now done this and found that this difference is indeed significant ($p = 0.0073$). We agree with the reviewer that if the errors shown represented the standard error or the 95% confidence interval then their overlap would rule out statistical significance, but as we instead show standard deviation this is not the case.

Supplementary Fig 9: differences presented here and described in the main text should be tested for statistical significance.

We have now performed statistical figures and added p values to this figure and for all figures where relevant. Fig. 4b-f, 5g-h, Supplementary Fig. 6a, 9a-d, and 12 have all been updated.

Minor comments:

Line 94 should read: including many *that* are stage-specific.

Thank you. This has been fixed.

Figure 3: Check panel numbering in legend – ‘e’ and ‘f’ refer to panels ‘d’ and ‘e’.

These two panels have now been moved to Supplementary Fig. 5 and the figure numbering throughout has been changed to be consistent

Line 203 refers to panel 3d which I believe is not presented in this version of the manuscript.

The original reference to 3d was correct, but this panel has now been moved to Supplementary Fig. 5 and the figure number changed.

Nature Communications Reviewer 2:

This manuscript by Josling *et al.* dissects the role of the AP2-G (previously identified master switch of the sexual commitment in the malaria parasites) in the regulation gene expression throughout the gametocyte generation process. The authors used the *Plasmodium falciparum* line in which C-terminally-tagged AP2-G is put under the control of the inducible expression system to synchronously induce gametocytogenesis and analyse the binding profile of AP2-G and the transcriptome at different time-points of this process. They also compared the AP2-G targets in the same cycle vs. next cycle committed parasites and use the cas9-mediated mutagenesis to confirm the previously postulated role of AP2-G indirectly regulating its own expression. Importantly the manuscript shows the unexpected interplay between AP2-G and another transcription factor (AP2-I), with both factors binding overlapping set of promoters at the schizont stage.

This paper represents a significant advancement in the field of gametocytogenesis and gene expression regulation in *Plasmodium*, providing first comprehensive list of the direct AP2-G targets and revealing the role this protein may play in the regulation of the invasion related genes. The main conclusions are clearly stated, backed-up by the data from multiple sources and the manuscript is of high interest to both the malaria researchers and to the parasitology community in general. As I was asked to provide my opinion regarding the comments of the previous reviewer rather than an independent review, I will start by addressing them.

We thank the reviewer for this overall positive assessment of the work presented in the manuscript. We are please to see that you feel this work is significant to our understanding of gametocyte commitment.

1. PfAP2-G regulates the expression of many early gametocyte genes. The PfAP2-G DD line has been used in Poran *et al.*, Nature 2017 and Kafsack *et al.*, Nature 2014. The first paper analyses global transcriptional differences using a single cell approach on synchronized parasites with time points extremely similar to this submitted paper. The second paper shows that PfAP2-G is upregulated in gametocyte producing lines and the subsequent transcriptional profile of knockdown show a cluster of downregulated genes is highly enriched for genes expressed during the first stages of gametocyte formation. In Line 118, the authors state “PfAP2-G is a direct activator of early gametocyte genes and ... the key driver of gametocytogenesis.” My argument would be that the Poran and Kafsack papers show this already.”

The causative link the AP2-G expression and gametocytogenesis has been indeed shown before in at least 6 manuscripts (Kafsack *et al.*, 2014, Poran *et al.*, 2017, Sinha *et al.*, 2014, Kent *et al.*, 2018, Bancells *et al.*, 2019, Brancucci *et al.*, 2018). The authors themselves are acknowledging this in the introduction, citing the relevant papers and providing a brief summary of the past findings. However, none of the previous papers could confidently discriminate between the direct targets of AP2-G and other genes, co-induced during the early gametocytogenesis by secondary factors. They were also unable to detect the genes, which, while regulated by AP2-G, were not showing strong changes in expression in the committed population. I would therefore agree with the author statement: “We have now identified direct targets of PfAP2-G and thus the earliest genes expressed during commitment, providing unprecedented insight into the events occurring during early gametocyte development.”. I believe that the current version of the manuscript addresses this point in a satisfactory way.

We thank the reviewer for recognizing the novelty of our work and how it complements previous studies that have laid the foundation for our study.

2. PfAP2-G regulates genes encoding proteins important for red blood cell invasion. While the authors show that PfAP2-G was associated with promoters of many genes in red blood cell invasion, only one gene *msrp1* shows a different expression profile with or without PfAP2-G. Furthermore, they observe that AP2-G and AP2-I bind to the promoters of these invasion genes. While the data

looks convincing that AP2-G and AP2-I may be binding as close or similar sites, the authors extend this observation to make a statement that the regulation of gametocytogenesis and invasion are linked during malaria parasite development. I am not fully convinced that the authors have enough mechanistic data to make that link. One potential experiment is to determine if the knockdown lines have a defect in invading reticulocytes using parasite growth assays and how this changes the dynamics of commitment. Having alternative biological assays looking at invasion could provide a more mechanistic link rather than a correlation of binding sites of particular cluster of genes.

I agree with the reviewer that one needs to be careful regarding the interpretation of the binding of AP2-G to the promoters of invasion genes. The lack of detectable expression changes in most of these genes combined with the fact that AP2-G expression in schizonts is not a pre-requisite for successful gametocytogenesis, suggests that its role at this stage is at best modulatory. While it is certainly tempting to speculate that committed merozoites exhibit different invasion patterns and AP2-G is responsible for it, at the moment there is little data to substantiate these claims. However, I don't believe that additional experiments tackling this question are necessary to complete the manuscript, as they would require a substantial amount of work, which would be out of the scope of the topic of this work – identification of the AP2-G targets. Authors should make clear throughout the manuscript that their interpretation of the role of AP2-G in schizonts is purely speculative and would require further in-depth studies.

We thank the reviewer for recognizing that determining the effect of modulation of invasion genes on committed merozoites is beyond the scope of this study. It will certainly serve as the foundation for future studies. We have also changed a sentence in the results to emphasise that our results related to invasion are not conclusive and require further validation: “The delay between peak binding of PfAP2-G and the down-regulation of invasion genes likely rules out a role for PfAP2-G as a direct repressor of invasion genes, though further studies will be necessary to determine the function of PfAP2-G binding to these promoters”

We have also added the following sentence to the Discussion: “Further work will be necessary to establish whether this model is correct or not and to establish the functional role of PfAP2-G at invasion gene promoters”.

Finally, we now describe PfAP2-G as a “potential” regulator of invasion genes rather than a regulator of invasion genes in the abstract and throughout the manuscript.

3. Motif identification for PfAP2-G and PfAP2-I and mechanism of binding. It would have been useful to see a more thorough examination of the putative motif site and data towards the independent/cooperative binding between these two transcription factors. Using EMSA as the Llinas lab has done before, they could interrogate the critical residues of the motif using mutated sites and also the sequence of binding (whether independent or at the same time).

The current version of the manuscript provides a good overview of the motif bound at different life stages by AP2-G (see Fig S2). I agree that it would be useful to compare the relative binding affinity of the AP2 domains of AP2-G and AP2-I to the consensus sequences enriched within the shared peaks and to each other in vitro motifs. As the binding specificity of both domains have been extensively studied by Llinas group using protein binding microarrays containing all possible variants of the 8bp DNA motifs (Campbell et al., 2010), perhaps the relevant information could be extracted from that dataset?

We thank the reviewer for this suggestion. We had also thought of this type of analysis due to the motif overlap. Unfortunately our original data from Campbell *et al.* is not very informative, because PfAP2-I AP2 domain 3 is a far more superior binder *in vitro* than the PfAP2-G AP2 domain. Therefore, although we can find some overlap in the suggested datasets, it is not

conclusive. We realized this some time ago, and have initiated a large number of new studies which use a next-generation protein binding microarray platform called a genomic context PBM (or gcPBM) (See: PMID: 23562153 and PMID: 29605182). This approach is tailored specifically to the genome of *Plasmodium falciparum* and contains thousands of sequences to query specificity. On the gcPBM all of the DNA probes come directly from intergenic regions present in the *P. falciparum* genome rather than random sequences. Therefore, we can now directly assay these two proteins on the same sequences they would (in theory) encounter within the nucleus. This work is ongoing and will be the subject of future manuscripts.

I also have two additional comments regarding the manuscript itself:

4. One of the more interesting findings of the manuscript is the fact that AP2-G appears to bind different sets of targets depending on the commitment pathway (NCC and SCC). However, besides noting this fact, authors provide little information regarding these differences. It would be helpful to see additional analysis including GO terms enriched in SSC- and NCC- exclusive gene sets, the consensus sequence motifs bound in the two datasets, and comparison with the previous AP2-G invasion profiles (eg. are genes bound exclusively in the NCC pathway the ones that are also bound in committed schizont, ring stages? Is AP2-G binding to its own promoter in both pathways? Is the MSRP1 binding present in SSC pathway even if these parasites won't pass through the merozoite stage?). That would presumably allow to generate some testable hypotheses regarding the differences in the AP2-G function in the two pathways.

We thank the reviewer for this suggestion. We have now added GO analyses of genes associated with NCC- and SCC-specific peaks (Supplementary Table 7), motifs enriched within NCC and SCC peaks (Supplementary Fig. 5b, c), and comparisons with binding sites in other stages (Supplementary Fig. 5d, Supplementary Table 7). We also now discuss binding in NCC and SCC to promoters of invasion genes, *ap2-g*, and *msrp1*.

We have added the following discussion of these analyses to the results: “As expected, most of the PfAP2-G binding sites identified in NCC cells and SCC cells were identified in stage I gametocytes in the previous set of experiments (Supplementary Table 7, Supplementary Fig. 4d). PfAP2-G was associated with its own promoter in both NCC and SCC cells. However, SCC-specific peaks were often found only in schizonts and sexual rings, whereas NCC-specific peaks were predominantly found only in stage I gametocytes. Gene Ontology analysis revealed that the genes associated with SCC-specific peaks were enriched in functions related to invasion (though this represented only four genes) and genes associated with NCC-specific peaks were most enriched in functions related to glucose catabolism (though again, this involved only three genes) (Supplementary Table 7). Though some binding to invasion gene promoters is seen in both NCC and SCC cells, fewer invasion gene promoters were bound than in schizonts and sexual rings. Most of the invasion genes bound in SCC cells are also bound in NCC cells, including *msrp1*.”

5. Gametocytes generated by Plasmodium can be either male or female. While the two populations cannot be morphologically separated till later in the development, it was previously speculated that the sex of gametocyte is predetermined during the commitment as all merozoites from the same schizonts would all differentiate into males or females (Silvestrini et al, 2000). Stage I gametocyte pool studied by the authors would therefore contain two separate populations. Are the genes induced in the stage I gametocytes by AP2-G expressed in both male and female gametocytes later in the infection, or some of them show strong sex-specific bias? Adding an extra column to one of the supplementary data tables, specifying this information (available from Lasonder et al., 2016) would answer this question and provide some information regarding the role of the AP2-G in sex determination.

We thank the reviewer for this suggestion. We have now added a comparison with this dataset to Supplementary Table 5 and discuss this in the results: “We found that PfAP2-G binds genes that are differentially transcribed in both male and female gametocytes with no clear bias in either direction, though genes that are ultimately expressed at the protein level are over-represented amongst PfAP2-G targets²¹”

Reviewers' Comments:

Reviewer #1:

Remarks to the Author:

The authors have addressed my comments and concerns appropriately and I am supportive of publication of this manuscript.

Reviewer #3:

Remarks to the Author:

I believe that all my comments have been addressed in a satisfactory way and I have no further concerns regarding the manuscript.

Response to Reviewers for NCOMMS-19-24067A

Nature Communications Reviewer #1:

The authors have addressed my comments and concerns appropriately and I am supportive of publication of this manuscript.

Nature Communications Reviewer #3:

I believe that all my comments have been addressed in a satisfactory way and I have no further concerns regarding the manuscript.

We thank the reviewers for their feedback and for improving our manuscript overall through their insightful comments and suggestions.